# Signalome-wide assessment of host cell response to hepatitis C virus

Gholamreza Haqshenas[1], Jianmin Wu[2,3,4], Kaylene J. Simpson[5,6], Roger J. Daly[7], Hans J. Netter[1,8], Thomas F. Baumert[9,10] & Christian Doerig[1]

Host cell signalling during infection with intracellular pathogens remains poorly understood. Here we report on the use of antibody microarray technology to detect variations in the expression levels and phosphorylation status of host cell signalling proteins during hepatitis C virus (HCV) replication. Following transfection with HCV RNA, the JNK and NF-κB pathways are suppressed, while the JAK/STAT5 pathway is activated; furthermore, components of the apoptosis and cell cycle control machineries are affected in the expression and/or phosphorylation status. RNAi-based hit validation identifies components of the JAK/STAT, NF-κB, MAPK and calcium-induced pathways as modulators of HCV replication. Selective chemical inhibition of one of the identified targets, the JNK activator kinase MAP4K2, does impair HCV replication. Thus this study provides a comprehensive picture of host cell pathway mobilization by HCV and uncovers potential therapeutic targets. The strategy of identifying targets for anti-infective intervention within the host cell signalome can be applied to any intracellular pathogen.

[1] Infection & Immunity Program, Monash Biomedicine Discovery Institute and Department of Microbiology, Monash University, Clayton Victoria 3800, Australia. [2] Kinghorn Cancer Centre & Cancer Division, Garvan Institute of Medical Research, Sydney, New South Wales 2010, Australia. [3] St Vincent's Clinical School, University of New South Wales, Sydney, New South Wales 2010, Australia. [4] Key laboratory of Carcinogenesis and Translational Research (Ministry of Education/Beijing), Centre for Cancer Bioinformatics, Peking University Cancer Hospital & Institute, Beijing 100142, China. [5] Victorian Centre for Functional Genomics, The Peter MacCallum Cancer Centre, St Andrews Place, East Melbourne, Victoria 3002, Australia. [6] Sir Peter MacCallum Department of Oncology, The University of Melbourne, Parkville, Victoria 3052, Australia. [7] Cancer Program, Monash Biomedicine Discovery Institute and Department of Biochemistry and Molecular Biology, Monash University, Clayton Victoria 3800, Australia. [8] Victorian Infectious Diseases Reference Laboratory, The Peter Doherty Institute, Melbourne Health, Victoria 3000, Australia. [9] Inserm U1110, Institut de Recherche sur les Maladies Virales et Hépatiques, Université de Strasbourg, 67091 Strasbourg, France. [10] Institut Hospitalo-Universitaire, Pôle Hépato-digestif, Hôpitaux Universitaires de Strasbourg, 67091 Strasbourg, France. Correspondence and requests for materials should be addressed to G.H. (email: gholamreza.haqshenas@monash.edu) or to C.D. (email: christian.doerig@monash.edu).

Hepatitis C virus (HCV), a member of the family *Flaviviridae*, infects approximately 2% of the world population. Infection leads to chronic hepatitis, which can progress to liver cirrhosis and hepatocellular carcinoma. Furthermore, several studies indicate a significant association with metabolic disorders, such as steatohepatitis, type II diabetes and insulin resistance[1–4]. The virion contains a positive sense single-stranded RNA genome, which comprises a single open reading frame translated into a large (∼3000 amino acids) polyprotein, which is enzymatically cleaved into at least 10 mature proteins: core protein, E1, E2, p7, NS2-3, NS4A-B, and NS5A-B[5]. HCV contains a structured non-coding region at the 5′ end of its genome that forms an internal ribosomal entry site and thus mediates viral genome translation (reviewed in ref. 6).

In recent years, the introduction of several direct acting agents, including small molecules targeting NS3, NS5A and NS5B, has revolutionized HCV therapy and led to sustained virological response (defined as aviremia 24 weeks after completion of antiviral therapy for chronic HCV infection) in the majority of treated patients[7,8]. However, direct acting agents' failure and resistance in a subset of patients, as well as limited or absent access to therapy for a large majority of patients worldwide, remain important challenges that need to be addressed by complementary therapies[9].

HCV, similar to other viruses, exploits host cell factors, structures and signalling pathways that control the host cell environment and are crucial for successful infection. Host cell factors required for viral replication represent attractive antiviral targets, since the most parsimonious way for the virus to develop resistance, that is, the selection of mutations in genes encoding those targets, will not be available. Several studies have explored the role of host cell signalling molecules in HCV replication. Global proteomic and analyses of liver biopsies[10,11] and HCV cell culture systems[11,12] have revealed significant changes in cellular environment after infection with HCV. Also, several reports investigated the role of host cell protein kinases as major signalling elements in HCV replication and assembly[13–19]. However, there are no reports of comprehensive studies focussing on time-dependent mobilization of host cell factors during the early steps of HCV replication.

We previously used the Kinexus antibody-based approach to investigate interactions between the malaria parasite *Plasmodium falciparum* and its host erythrocyte and demonstrated that a signalling pathway implicating the human kinases p21-activated kinase and mitogen-activated protein kinase (MAPK) kinase 1 is activated by infection[20]. Here we report that implementation of the Kinexus antibody microarray technology coupled to functional validation of hits by small interfering RNA (siRNA) confirmed a number of host cell factors, notably protein kinases, that were previously identified as modulators of HCV infection; this represents a useful positive control for our new approach. Importantly, this also revealed several novel host cell signalling pathways that are mobilized by HCV. We provide evidence that treatment of infected cells with a selective chemical inhibitor of MAP4K2, one of the protein kinases found to be activated by infection, severely affects HCV genome replication. This constitutes a proof-of-concept that this system-wide approach can deliver novel targets for antiviral intervention.

## Results

### Signalling pathways affected by viral genome transfection.
Replication-competent HCV RNA was transfected into the hepatocyte-derived Huh7.5.1 cell line, and transfection efficacy was verified by immunofluorescence assay using an anti-NS5A antibody (Supplementary Fig. 1). Alterations in host cell signalling pathways were investigated 6, 12 and 24 h after transfection using the Kinexus antibody microarray (Fig. 1a; see Methods section for a description of the array). This allowed us to quantify the protein expression levels of cell signalling factors, as well as phosphorylation site occupancy on many of these factors, that were upregulated or downregulated at various time points post-transfection. The full and short-listed (103 genes) inventories of factors modulated at the 6, 12 and 24 h time points are shown in Supplementary Data 1–3, respectively. A bioinformatic analysis of the data sets, performed to test statistical enrichment of KEGG pathways (see Methods section), revealed that multiple signalling pathways were modulated (Supplementary Data 4), including the nuclear factor (NF)-κB, signal transducer and activator of transcription/Janus kinase (STAT/JAK) and MAPK pathways, as well as calcium signalling and components of the cell cycle progression and apoptosis machineries. The number of cell factors whose expression or phosphorylation status was modulated by infection generally increased over time; however, the number of downregulated genes was lower at 24 h than at 12 h post-transfection (Fig. 1b,c). No signalling molecules were observed to be modulated at all three time points (Fig. 1d). A heat map was generated to facilitate the comparison of the levels of proteins (or their phosphorylation status) across the three time points (Supplementary Fig. 2).

To determine whether the 103 cell factors identified as modulated by infection in the microarray experiment had a role in HCV replication, siRNA gene silencing was used as summarized in Supplementary Fig. 3; the 103 genes and their specific SMARTpool siRNA sequences are presented in Supplementary Data 5. The modulation of HCV replication following silencing of cell factors was measured using a reporter system based on secreted luciferase enzyme encoded by a modified viral genome, which is an accepted quantifier of virus replication[21–23]. The results were normalized against cell viability and are presented in Supplementary Data 6. The primary screen was validated by direct, quantitative reverse-transcription polymerase chain reaction (qRT–PCR) quantification of intracellular viral genomic RNA levels following infection with the virus; in this case, the results were normalized against mRNA levels of the housekeeping genes *β-actin* and *GAPDH* and are presented in Supplementary Data 7. The combination of the two systems provided strong evidence for the implication of several cell factors in HCV replication. For a few tested factors, some discrepancy between the results of the luciferase and qRT–PCR readout systems was observed (see Discussion section).

The top 10 factors that play a promoting or inhibiting role in HCV replication are presented in Fig. 1e, and the full list of the 103 cell factors tested in RNAi experiments can be found in Supplementary Data 6. The sections below focus on the JAK/STAT, NF-κB, calcium and MAPK signalling pathways that our microarray and gene silencing analyses showed to be important in HCV replication. Data pertaining to several additional signalling pathways, namely, the insulin/phosphoinositide-3 kinase/AKT and Wnt/β-catenin pathways, as well as components of the machinery regulating cell cycle progression and apoptosis, are discussed in Supplementary Note 1, with the relevant data presented in Supplementary Figs 7–10.

**JAK/STAT5A pathway.** JAK/STAT signalling integrates extracellular environment signals through a complex network. Different STAT transcription factors are activated by various upstream signals: for example, activation of STAT1 and STAT2 is initiated by binding of interferon to its receptor, while STAT5A is activated by interleukin-2 (ref. 24). Upon activation by JAK2-dependent phosphorylation[24], STAT5A is translocated into the nucleus where it promotes transcription of target genes, including

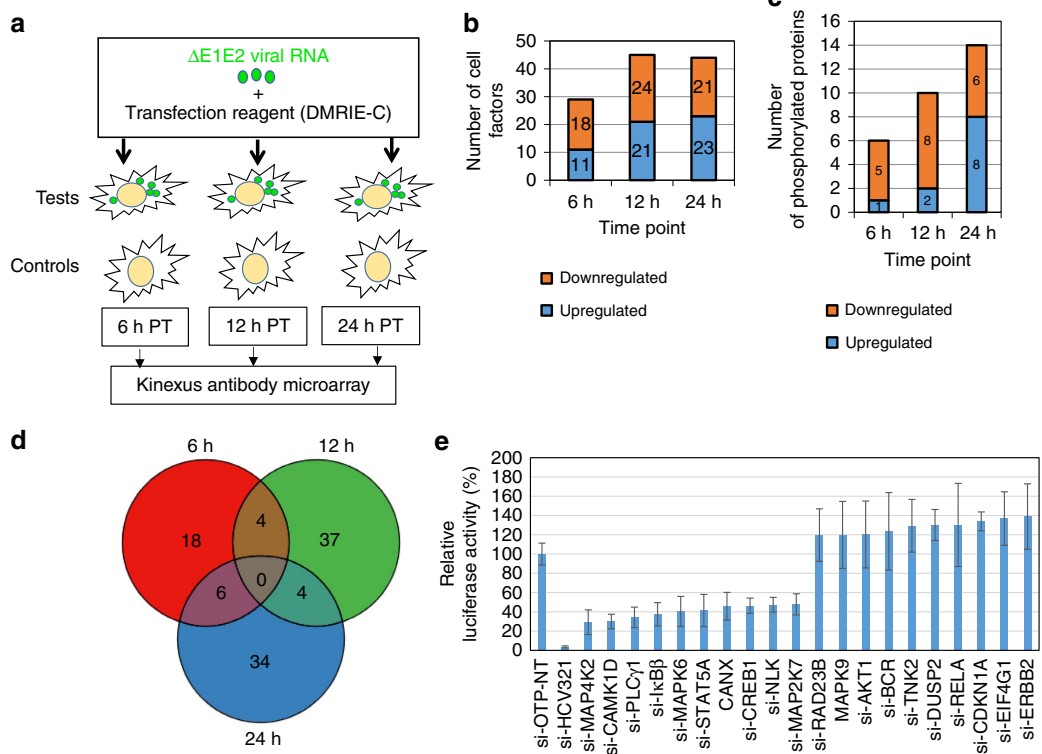

**Figure 1 | Flowchart of the Kinexus antibody microarray methodology.** (**a**) *In vitro* synthesized transcripts of HCV deletion mutant (ΔE1E2) were transfected into Huh7.5.1 cells using transfection lipid reagent DMRIE-C. Negative control cells (Controls) were transfected with the transfection mixture lacking viral genome. At given time points post-transfection (PT), cells were harvested and lysed, and the extracts were incubated onto the antibody microarray. (**b,c**) Number of total and phosphorylated forms of signalling molecules across three time points. (**d**) Venn diagram of the 103 identified genes across different time points (6, 12 and 24 h post-transfection) (see Supplementary Data 1–3). (**e**) Effects of silencing of the genes identified by antibody microarray on virus replication. The graph shows the top 20 factors that had most profound effect on virus replication. Following silencing of the genes for 43 h, cells were infected with a reporter virus containing a Renilla luciferase gene. The luciferase activity of each well was measured and normalized to its viability and negative control si-OTP-NT set at 100%. The error bars represent s.d. of at least three wells. *P* values were calculated with the unequal variance *t*-test embedded in Excel; significant variations ($P < 0.05$) are depicted by an asterisk.

B-cell lymphoma—extra large (Bcl-xl)[25] and c-myc[26]. It has been reported that HCV regulates JAK/STAT pathways, notably through degradation of STAT1 and STAT3 and induction of SOCS3, a member of the suppressors of cytokine signalling (SOCS) family that inhibits STAT phosphorylation by binding and inhibiting JAKs[27,28]. Our antibody microarray data reveal that the phosphorylated forms of both JAK2 and STAT5A are increased 24 h post-transfection (Supplementary Data 3, Fig. 2), suggesting a role for the JAK/STAT5 signalling pathway in virus replication. These results are consistent with clinical findings that STAT5A is upregulated by HCV infection[29]. Unexpectedly, and as discussed below in the context of the NF-κB pathway, Bcl-xl showed a decrease at 24 h and c-myc level did not show significant changes, suggesting that STAT5A may not target these genes in the context of HCV infection. Nonetheless, STAT5A clearly plays an important role, because silencing of either JAK2 or STAT5A resulted in a reduction in virus replication; the effect of STAT5A silencing was equally strong when assessed by luciferase or qRT–PCR, while the effect of JAK2 silencing was more pronounced when assessed by luciferase assay than by qRT–PCR (see Discussion section; Fig. 2 and Fig. 1e; Supplementary Data 6). It is established that STAT5 promotes replication of human papillomavirus[30] and polyomavirus[31]; that it also promotes replication of an RNA virus reveals that this strategy is widespread among animal viruses.

**NF-κB pathway**. The NF-κB transcription factor regulates multiple cellular processes, including immune response, inflammation, apoptosis, autophagy, cell adhesion, differentiation and proliferation[32]. There are two distinct NF-κB pathways, the canonical and non-canonical[33]. The classical pathway comprises NF-κB1 (p50), RelA (p65), the inhibitors of NF-κB IκBα and IκBβ and their cognate kinases IκB kinases α and β (IKKα, IKKβ). The non-canonical pathway is regulated mainly by the NF-κB-inducer kinase (NIK, also known as MAP3K14), IKKα, IκBα, p55 and RelB. Major alterations in these pathways were observed 12 h post-transfection with HCV genomic RNA. The levels of both NF-κB1(p50) and RelA (p65) were reduced 12 h post-transfection (Supplementary Data 2, Fig. 3), confirming the previously reported suppression of the canonical NF-κB pathway by HCV[34]. Expression of NF-κB-dependent anti-apoptotic proteins, such as Bcl-xL, X-linked inhibitor of apoptosis protein and the long form of cellular-FLICE inhibitory protein, was decreased after infection of cultured cells, in line with the reported decreased levels of Bcl-xL, X-linked inhibitor of apoptosis protein and cellular-FLICE inhibitory protein (mRNA and protein) in livers with chronic hepatitis C[34]. siRNA-mediated silencing of the NF-κB inhibitors IκBα and IκBβ reduced virus replication, but in contrast, and consistent with the published results[35], silencing RelA promoted virus replication (Supplementary Data 6, Fig. 3); RelA supresses HCV replication through a RelA-dependent β-interferon production

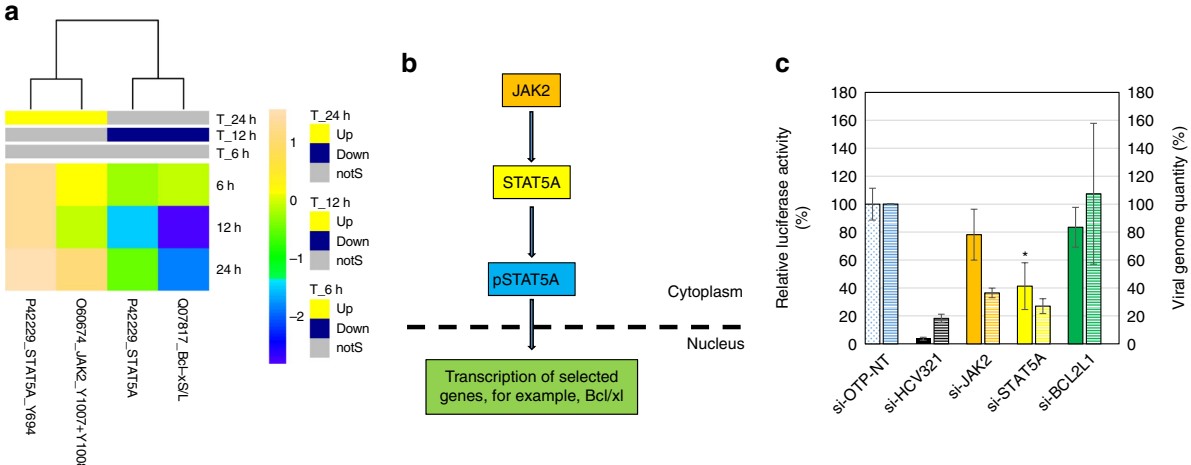

**Figure 2 | JAK/STAT5A pathway.** (**a**) Upregulation and downregulation of cell factors determined by antibody microarray experiment. Factors with a $Z$ ratio $>1.25$ are considered significant (see Methods section). (**b**) A schematic figure of simplified JAK/STAT5A pathway. Colour boxes represent factors shortlisted for siRNA validation following the microarray experiment. (**c**) Effect of silencing of the JAK/STAT5A pathway components on HCV replication. Following silencing the target genes for 43 h, cells were infected with a reporter virus containing a Renilla luciferase gene. Solid bars, luciferase readout: Luciferase activity in each well, normalized to viability and negative control (si-OTP-NT) set at 100%. Hatched bars, qRT–PCR readout, normalized to two housekeeping genes. The error bars represent s.d. of at least three wells. $P$ values were calculated with the unequal variance $t$-test embedded in Excel; significant variations ($P<0.05$) are depicted by an asterisk.

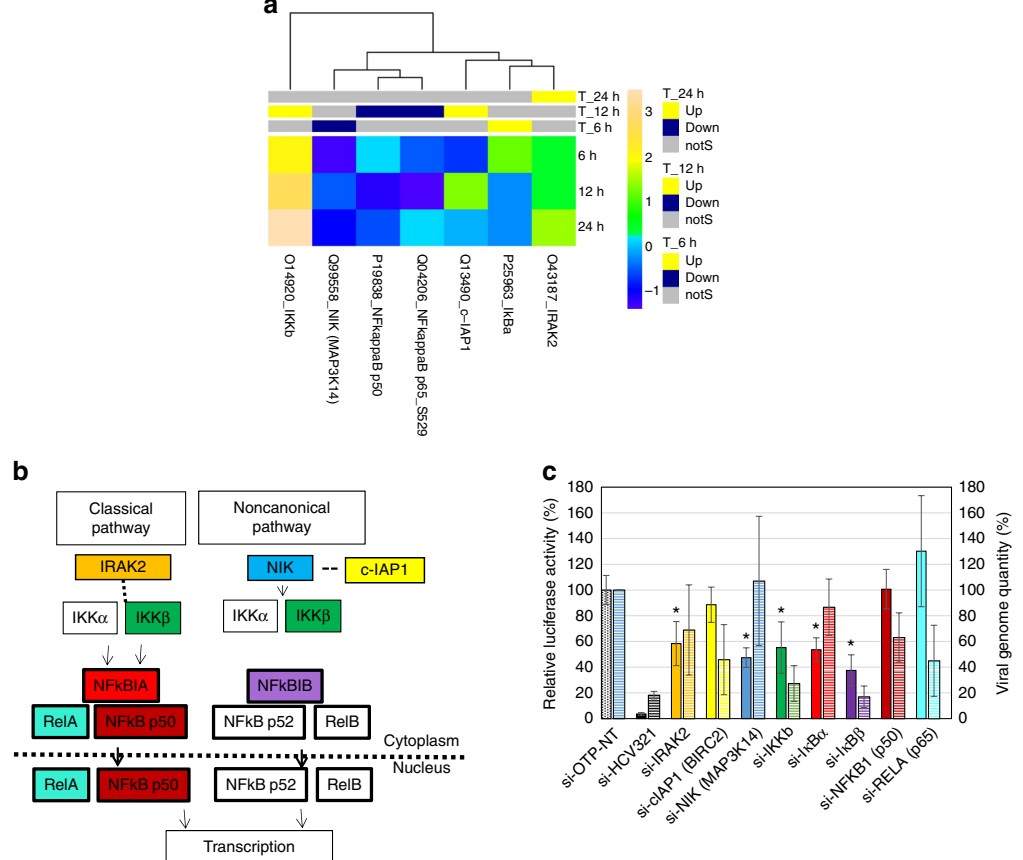

**Figure 3 | Non-canonical NF-κB pathway.** (**a**) Upregulation and downregulation of cell factors determined by antibody microarray experiment. Factors with a $Z$ ratio $>1.25$ are considered significant (see Methods section). (**b**) A schematic figure of the (simplified) NF-κB pathway. Colour boxes represent factors shortlisted for siRNA validation following the microarray experiment. (**c**) Effect of silencing of the NF-κB pathway components on HCV replication. Following silencing the target genes for 43 h, cells were infected with a reporter virus containing a Renilla luciferase gene. Solid bars, luciferase readout: Luciferase activity in each well, normalized to viability and negative control (si-OTP-NT) set at 100%. Hatched bars, qRT–PCR readout, normalized to two housekeeping genes. The error bars represent s.d. of three independent experiments. $P$ values were calculated with the unequal variance $t$-test embedded in Excel; significant variations ($P<0.05$) are depicted by an asterisk. IκBα and IκBβ are referred to as NFKBIA and NFKBIB in the Kinexus microarray data set.

mechanism[36]. There was some discrepancy between the luciferase and qRT–PCR readouts for RelA and NIK, but both assays concur in a strong impairment of viral replication when IκBα and IKKβ are silenced. Taken together, these data clearly indicate that activation of the canonical NF-κB pathway suppresses virus replication. In the non-canonical NF-κB pathway, interleukin-1 receptor-associated kinase 2 (IRAK2) recruits the tumour necrosis factor-α receptor-associated factor 6 and NIK, an activator of IKKα[37]. We detected an increase in IRAK2 levels at 24 h but a decrease in NIK at 6 h (Supplementary Data 1–3). Since NIK is constitutively expressed in its active form, abundance of the protein reflects activation of the pathway[33]. Interestingly, silencing IRAK2 and NIK reduced virus replication as indicated by lower luciferase activity (Fig. 3, Supplementary Data 6), suggesting an infection-promoting role for these two kinases. NIK plays a key role in the replication of several viruses, such as respiratory syncytial virus, Epstein–Barr virus and Herpesvirus saimiri, through activation of the NF-κB pathway. NIK phosphorylates IKKα, which in turn (independently from downstream elements of the NF-κB pathway) promotes HCV assembly[35]. NIK is downregulated 6 h post-transfection and never reaches the same level as the mock-infected cells control throughout the duration of the experiment (12 and 24 h; Supplementary Data 1–3). IKKα, the substrate of NIK, phosphorylates and destabilizes NIK in a negative feedback loop. Phosphorylation of IKKα increases during HCV infection[35], which may explain the lower quantities of NIK we observed. Interestingly, silencing NIK, similar to IKKα silencing[35], remarkably reduced HCV replication (see above). Unexpectedly, silencing cellular inhibitor of apoptosis 1, which destabilizes NIK by promoting its ubiquitination[38,39] and was increased at 12 h, also reduced virus replication (more noticeable when viral genome quantity was measured, Fig. 3). Our data suggest that NIK and IKKβ play essential positive roles in HCV replication, as previously reported for IKKα[35]. To our knowledge, this is the first report suggesting a promoting role for NIK in a viral infection.

IRAK2 is activated by interleukin 1, a cytokine that has an inhibitory effect on HCV replication, as suggested through the use of a replicon system[40]. In contrast, our results indicate that IRAK2 has a promoting role in HCV replication when the luciferase-based assay was used to measure infectivity (Fig. 3, Supplementary Data 6). This discrepancy might be due to the use of different systems, that is, replicon versus cell culture.

**Calcium signalling.** Calcium influx and its release from cellular stores are important for replication of several viruses[41–44]. One of the key regulators of calcium homeostasis is phospholipase C γ1 (PLCγ1), which is activated by tyrosine kinases such as epidermal growth factor receptor[45]. Upon activation, PLCγ1 hydrolyses phosphatidylinositol 4,5-bisphosphate and produces the second messengers inositol triphosphates and diacylglycerol[45]. Inositol triphosphate is oligomerized and inserted into the endoplasmic reticulum and mitochondrial membranes, causing calcium release from these cellular compartments. The released calcium mediates activation of several protein kinases, including calmodulin-dependent kinase (CaMK), nemo-like kinase and focal adhesion kinase (FAK). CaMK induces the G1–S phase transition in cell cycle progression and is utilized by several viruses for promoting their replication[46,47]. Our microarray experiment revealed that both epidermal growth factor receptor and PLCγ1 are downregulated 12 h post-transfection (Supplementary Data 1). Using both detection systems, PLCγ1 silencing reduced virus replication (Supplementary Data 6 and 7, Fig. 4), suggesting a promoting role for this cell factor in HCV replication.

Furthermore, CaMK1D suppression was observed at 24 h (Supplementary Data 3). However, the phosphorylated, active form of cAMP responsive element-binding protein 1 (CREB1), a substrate of CaMK[48], increased at 24 h post-transfection, suggesting the activation of CaMK at 24 h. Due to the absence in the array of antibodies detecting phosphorylated sites on CaMK1D and CaMK kinase, we were unable to assess the activation status of these enzymes; this deserves further investigations. As mentioned above, silencing CREB1 decreased virus replication, indicating a promoting role for this factor in virus replication; likewise, silencing CaMK1D suppressed virus replication as evidenced by the detection of both viral genome and luciferase activity. Another calcium-dependent kinase, FAK, was phosphorylated and activated at 24 h. Silencing FAK did not alter viral replication. Proline-rich tyrosine kinase 2 (Pyk2, also known as PTK2B), activated by CaMK1D, may activate the extracellular signal-regulated kinase (ERK) pathway. Diacylglycerol (see above) activates protein kinase C (PKC), which then phosphorylates cell factors such as MAPKs and IκB, through which it regulates several cellular functions[49]. PKCα and PKCε increased at 6 and 24 h post-transfection, respectively, but PKCε silencing reduced virus replication more than PKCα. PKCθ and PKCδ levels were lower than those of the controls at 6 and 12 h, respectively, and their silencing reduced virus replication as evidenced by both detection assays (Fig. 4).

**MAPK signalling.** MAPKs, which include the ERK, c-Jun N-terminal kinases (JNK) and p38, regulate major cellular functions, including proliferation, differentiation, development, innate immunity and apoptosis[50,51]. Our microarray data revealed a major suppression of several components of the JNK pathway (MAP4K2, MAP3K5, MAP2K7 and MAP2K4), with the exception of JNK2 (also known as MAPK9), which was increased at 24 h (Supplementary Data 3, Fig. 5); the other two Jun kinases, JNK1 and JNK3, were not significantly affected by infection. Silencing of most of the above kinases reduced virus replication (Supplementary Data 6, Fig. 5). Consistent with our microarray results, MAP2K7 has been previously reported to be downregulated by HCV NS5A through an unknown mechanism[52]. MAP4K2 is involved in activation of MAP2K4 and MAP2K7, which in turn phosphorylate and activate JNK. In contrast to the suppressive effects of MAP4K2 silencing on virus replication, silencing MAP2K4 did not significantly affect virus replication as evidenced by both detection systems used. However, silencing MAP2K7 reduced virus replication to a lower extent than silencing MAP4K2. The JNK pathway is stimulated by pathogen-associated molecular pattern and plays a role in the subsequent innate immune response[51]. Our data indicate that MAP4K2 and MAP2K7 play a role in HCV replication in a JNK pathway-independent manner, as suggested by the observation that the phosphorylation levels of the effector Jun protein did not alter throughout the study. Silencing JNK2 (MAPK9), whose levels increased at 24 h, reduced HCV replication remarkably when viral RNA genome was quantified, but the luciferase readout gave a lower effect.

Phosphatase of activated cells 1 (PAC1), also known as dual specific phosphatase 2, negatively regulates the MAPKs that are involved in cell proliferation and is a transcriptional target of p53. It has been reported that silencing PAC1 leads to the suppression of apoptosis and promotes cell survival[53]. Thus, in view of our data outlined above, it is not surprising that an increase in virus replication was observed upon silencing PAC1. MAPK6 (also known as ERK3), which is supressed at 24 h, was detected as one of the top 10 cell factors modulating HCV replication (Fig. 1e).

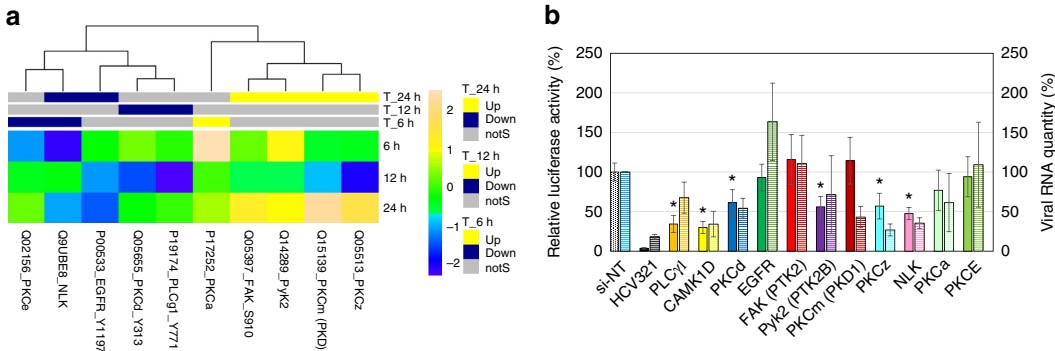

**Figure 4 | Calcium-dependent signalling. (a)** Upregulation and downregulation of cell factors determined by antibody microarray experiment. Factors with a $Z$ ratio >1.25 are considered significant (see Methods section). (**b**) Effect of gene silencing on HCV replication. Following silencing the target genes for 43 h, cells were infected with a reporter virus containing a Renilla luciferase gene. Solid bars, luciferase readout: Luciferase activity in each well, normalized to viability and negative control (si-OTP-NT) set at 100%. Hatched bars, qRT–PCR readout, normalized to two housekeeping genes. The error bars represent s.d. of at least three wells. $P$ values were calculated with the unequal variance $t$-test embedded in Excel; significant variations ($P<0.05$) are depicted by an asterisk.

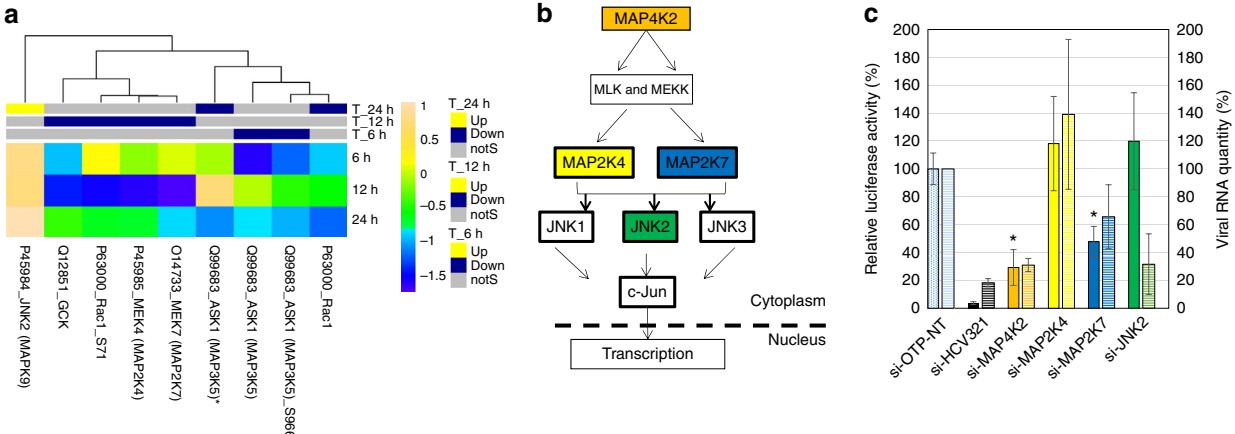

**Figure 5 | MAPK pathways. (a)** Upregulation and downregulation of cell factors determined by antibody microarray experiment. Factors with a $Z$ ratio >1.25 are considered significant (see Methods section). (**b**) A schematic figure of the simplified MAPK pathway. Colour boxes represent factors shortlisted for siRNA validation following the microarray experiment. (**c**) Effect of silencing of the MAPK pathway components on HCV replication. Following silencing the target genes for 43 h, cells were infected with a reporter virus containing a Renilla luciferase gene. Solid bars, luciferase readout: Luciferase activity in each well, normalized to viability and negative control (si-OTP-NT) set at 100%. Hatched bars, qRT–PCR readout, normalized to two house keeping genes. The error bars represent s.d. of three independent experiments. $P$ values were calculated with the unequal variance $t$-test embedded in Excel; significant variations ($P<0.05$) are depicted by an asterisk.

**Chemical inhibition of MAP4K2 impairs HCV replication.** As mentioned above, MAP4K2 silencing with SMARTpool siRNA reduced HCV replication, suggesting an important role for this kinase in the process. The SMARTpool results were verified by silencing experiments using individual siRNAs, with a non-targeting siRNA (si-OTP-NT) acting as negative control. The pool and all individual siRNAs reduced virus replication, albeit at different levels, and the individual siRNA 2 showed some toxicity in uninfected cells, presumably through off-target silencing (Fig. 6a,b). The effects of the expression knockdown of MAP4K2 on virus replication was evidenced by the two detection systems when three out of the four individual siRNAs were used (Fig. 6a,b). Western blot analysis demonstrated that both the SMARTpool and individual siRNA reduced MAP4K2 protein levels (Supplementary Fig. 4). This consolidates the evidence that MAP4K2 plays an important role in HCV replication and prompted us to test whether chemical inhibition of the enzyme would impair virus replication. Quite timely, a type II kinase inhibitor that displays selectivity for MAP4K2, TL4-12, has been recently developed[54]. The compound was non-toxic for Huh7.5.1

cells at 12 μM and lower concentrations (Supplementary Fig. 6), and we used a range of concentrations of TL4-12 (0.4–6 μM) to pretreat cells for 5 h prior to infection with a reporter virus expressing luciferase. Luciferase activity of the supernatant fluids was measured 12 and 36 h post-infection and normalized to the mock-treated (dimethyl sulfoxide (DMSO) vehicle only, no TL4-12) HCV-infected cells. TL4-12 treatment reduced luciferase activity levels (Fig. 6c) and RNA replication (Fig. 6d), suggesting an inhibitory effect on virus replication in a dose-dependent manner, whereas DMSO alone did not affect virus replication (we used the Graphpad program to calculate that the concentrations causing 50% toxicity and 50% HCV replication inhibition were 26.8 and 3.9 μM, respectively, giving an approximate therapeutic index of 6.8). Some kinase inhibitors directly inhibit luciferase activity[55]; to rule out that this was the case for TL4-12, cells were infected with virus for 24 h and analysed by western blotting using a monoclonal antibody to viral NS5A. The results demonstrated a significant reduction in HCV NS5A by treatment with 6 μM TL4-12 (Supplementary Fig. 5A,B). To provide further evidence that TL4-12 impaired

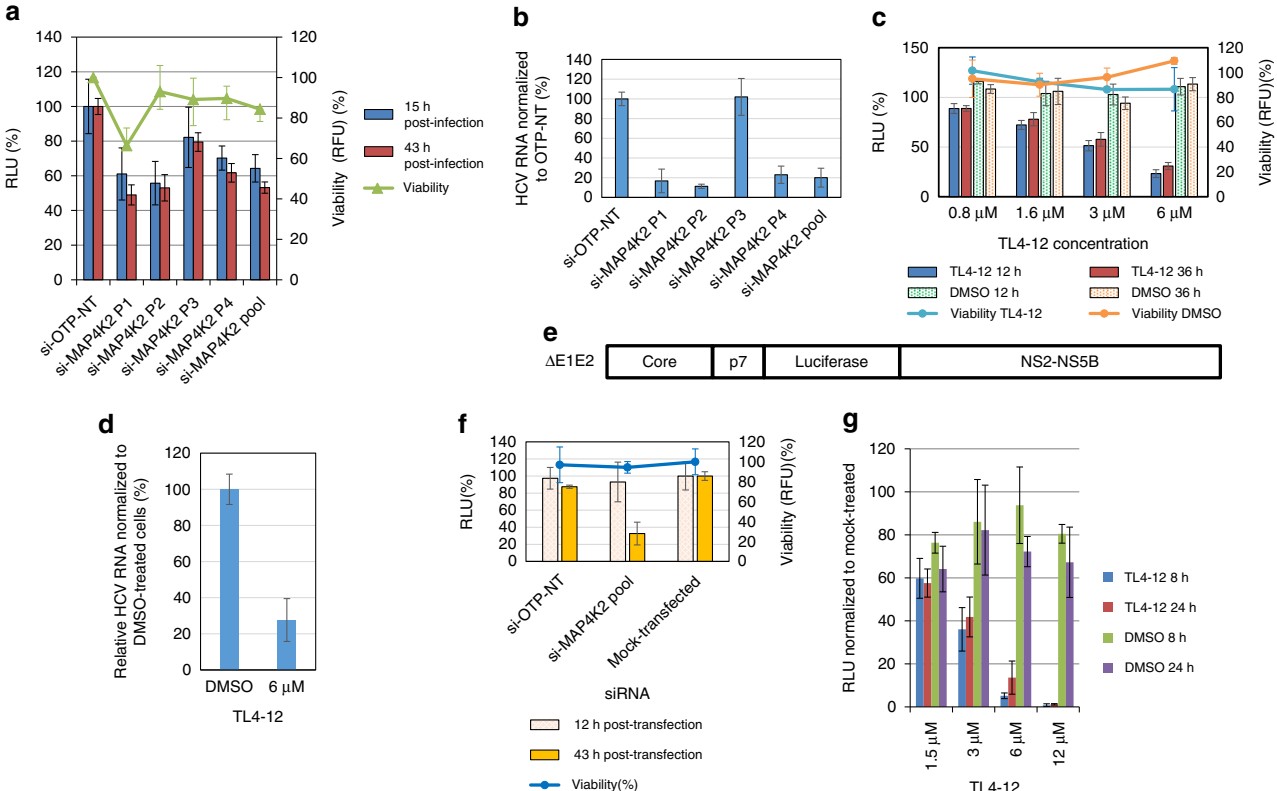

**Figure 6 | Effects of MAP4K2 silencing and chemical inhibition on HCV replication.** Huh7.5.1 cells were infected with a reporter virus after silencing MAP4K2 using SMARTpool and individual siRNA (si-MAP4K2 P1-4). The luciferase activity of (**a**) supernatant fluids was measured 15 and 43 h post-infection and (**b**) viral RNA quantity at 72 h post-infection. Cell viability was assessed using the PrestoBlue reagent (see Methods section) after the 43 h samples were collected. (**c**) Effects of TL4-12 on HCV replication. Huh7.5.1 cells were treated with TL4-12 for 5 h before the virus inoculum was added. At 6 h post-infection, cells were washed and the fresh medium containing TL4-12 were added. Samples were collected at given time points post-infection. DMSO-treated samples acted as mock-treated cells. The luciferase activity of (**c**) TL4-12 and mock-treated cultures and (**d**) viral RNA levels were normalized to non-treated cells. The viability was assessed using the PrestoBlue reagent. (**e**) Schematic representation of HCV replicon (Jc1ΔE1E2) lacking E1 and E2 genes. (**f**) Effect of MAP4K2 silencing on Jc1ΔE1E2 replication. (**g**) Effect of TL4-12 on Jc1ΔE1E2 replication. The error bars represent s.d. of three replicates. RLU: relative luciferase units; RFU: relative fluorescence units.

viral genome replication, the expression of viral protein NS5A was also tested by immunofluorescence assay. As shown in Supplementary Fig. 5C, the intensity of NS5A immunofluorescence signal in cells treated with TL4-12 was reduced, suggesting the inhibition of virus replication.

We then wanted to determine whether downregulation of MAP4K2 affects HCV genome replication. Using a reporter HCV replicon lacking the sequences encoding the major envelope proteins (E1 and E2) (ΔE1E2) (Fig. 6e), we demonstrated that MAP4K2 silencing and inhibition by TL4-12 interfere with virus replication at a post-entry stage (Fig. 6f,g).

Additional data pertaining to other signalling pathways are presented in Supplementary Note 1.

## Discussion

We utilized the Kinexus antibody microarray to generate a comprehensive picture of host cell signalling pathways that are modulated by HCV infection. Cellular processes regulated by key signalling cascades that are modulated during the early steps of replication include: (i) metabolism (insulin/AKT/phosphoinositide-3 kinase pathway), (ii) inflammation (NF-κB pathway), (iii) apoptosis, and (iv) cell proliferation (MAPK pathways, cell cycle control elements, calcium and Wnt signalling). Gene silencing experiments demonstrated that specific components of the signalling pathway identified in the microarray experiment are essential for HCV replication, while others play an inhibitory

role. We assessed the effect of gene silencing on viral replication in two independent sets of experiments, based on two different readouts: measurement of secreted luciferase encoded by a modified virus[21–23], and direct measurement of viral RNA by qRT–PCR. In many instances, the data obtained with each system overlap, while in others (for example, RelA and JNK2), considerable discrepancy is observed. This is likely due to the fact that silencing signalling genes may well interfere, either with the trafficking systems underlying luciferase secretion or with the expression of the housekeeping genes used as a normalization component in the qRT–PCR experiments. Nevertheless, those genes whose silencing give the same results with both readout methods are very strong candidates as regulators of viral replication. Interestingly, the major signalling components that promote HCV replication (CaMK1D, MAP4K2, PLCγ1) were downregulated at early time points after replication initiation, which may provide some insight into the process whereby limited HCV replication leads to chronic infection. Several novel promoting factors such as CaMK1D, MAP4K2, PLCγ1 and STAT5A were identified and clearly deserve further investigation. We also identified novel HCV-induced alterations in the insulin-induced pathway, whose characterization may unravel unidentified mechanisms of HCV pathogenesis, notably in the context of the association of this virus with type II diabetes.

Using a combination of RNAi and chemical inhibition, we demonstrated that the proximal activator MAP4K2 plays a role in

HCV genome replication. MAPK pathways are involved in several cellular physiological functions and are crucial to the life cycle of some viruses[56,57]. Several protein kinases have been demonstrated to mediate HCV entry, genome replication and translation[58,59]. While previous reports indicate that MAP4K2 plays a role in HCV entry[13], our data indicate that MAP4K2 is also mobilized for HCV genome replication. We demonstrated that a selective type II inhibitor of MAP4K2 kinase, TL4-12 (ref. 54), impairs virus replication, suggesting that inhibitors of this class have therapeutical potential for HCV patients. Furthermore, the kinase may have a role not only in virus replication but also in the transformation of infected cells; MAP4K2 chemical inhibition may thus kill two birds with one stone.

We further validated our microarray data by examining the data set from a published transcriptomics study performed by oligonucleotide microarray at early time points post-infection with HCV (6, 12 and 24 h)[60]. We analysed the data sets from this report using TFactS[61], a program designed to predict which transcription factors are activated or inhibited based on the expression of their target genes in a microarray data set. The output of the TFactS analysis of the published transcriptomics data set[60] is presented in Supplementary Data 8; overall, the host cell transcription factors that are predicted to be activated or inhibited by HCV infection fit well with our antibody microarray data: at early time points of HCV infection, only a limited number transcription factors is predicted to be modulated. Our data concur with the transcriptomics results in detecting activation of β-catenin at 6 h post-infection and CREB1 (at 12 h for the antibody microarray data, and 24 h for the transcriptomics data). Additionally, consistent with our microarray analysis, CREB1 and STAT5A were listed by the TFactS program as activated transcription factors at 24 h (Supplementary Data 8). SMAD1 was inhibited at 24 h in both studies (the decrease was slight in our data set). The Kinexus microarray did not contain antibodies against the activated form of several transcription factors (including SREBF1-2, CREBBP and SMAD3) that our TFactS analysis of transcriptomics data identified as modulated by infection. Nevertheless, the TFactS analysis suggests that STAT3 is activated at 12 h post-infection, while the activated form of STAT3 appeared at 6 h in the antibody microarray; it can actually be expected that the transcriptional response is in full swing at a later stage than the activation of the signalling pathway triggering the response. Surprisingly, the transcriptomics analysis[60] showed that the expression of some genes promoting apoptosis, and that of other genes promoting cell survival, were increased upon HCV infection; such an apparently opposite response was also observed with respect to cell cycle genes. This is in line with our antibody microarray data suggesting (i) that levels of Bax (a pro-apoptotic molecule) increased at 6 h post-transfection but decreased at 12 h, and (ii) that Bak (another pro-apoptotic factor) decreased in abundance at 24 h post-transfection. This would be consistent with the cell engaging an apoptotic response immediately after infection and with the apoptotic machinery being skewed towards cell survival at later stages of infection. Thus conflicting transcriptional changes revealed by the transcriptomics analysis appear to occur in response to HCV[60], which, in some instances such as that of the apoptotic response mentioned above, may reflect temporal regulation of opposite responses. This is in line with the complexity of the signalling pathway mobilization revealed by our study.

The combination of antibody microarray and gene silencing approaches allowed us to identify several factors that play an important role in the viral replication cycle. While some of these factors had previously been reported as influencing HCV

replication, several others are novel to our understanding of the molecular aspects of HCV infection. The combination of antibody microarray and gene silencing represents a novel, system-wide approach for the understanding of infectious disease biology and can be expected to provide a strong basis for the development of urgently needed lead compounds against not only viruses but also any intracellular pathogens of bacterial or protist taxons.

## Methods

**Constructs and cells.** Human hepatoma cell lines Huh7 and Huh7.5.1 (refs 61–64) were received from Professor Francis Chisari (Scripps, CA). Cells were cultivated[62,63] and rountinely tested for mycoplasma. The Jc1FLAG2(p7-nsGluc2a) construct[65], referred to as Jc1-Luc, was provided by Professor Charles Rice (Rockefeller University, CA). Overlapping PCR was used to delete the entire envelope glycoproteins E1 and E1 genes from the Jc1-Luc plasmid to yield the final construct named Jc1ΔE1E2.

**Viral RNA synthesis and transfection.** HCV cDNA constructs were purified (Qiagen Maxi Prep Kit), linearized with XbaI and re-purified by phenol-chloroform extraction. The Megascript T7 Transcription kit (Gibco-Ambion) was used to generate HCV transcripts in vitro, which were purified using phenol–chloroform extraction and stored at −70 °C. The DMRIE-C transfection reagent (Life Technologies) was used to introduce the transcripts into cells (4 μg RNA and 6 μl DMRIE-C for each well in a six-well plate).

**Immunofluorescence assay.** To assess the expression of viral proteins, the anti-NS5A monoclonal 9E10 (ref. 66) was kindly provided by Professor Charles Rice (Rockefeller University, CA). Cells were fixed with paraformaldehyde 3.5% for 1 h and permeabilized by 0.1% Triton-X100 in PBS. After blocking with 5% foetal bovine serum (FBS), cells were incubated with the NS5A antibody (1/10,000) and stained with chicken anti-mouse Alexa 488 (Invitrogen, Cat. no. A21200). The nuclei were stained with 4,6-diamidino-2-phenylindole for 5 min.

**Antibody microarray.** The Kinexus antibody microarray is an extremely sensitive approach allowing the detection of low-abundance signalling proteins. The microarray comprises ∼510 pan-specific antibodies against a large number of components of signalling pathways (to monitor protein expression) and ∼340 phosphosite-specific antibodies within those proteins (to monitor phosphorylation status). The screen encompasses 189 protein kinases, 31 protein phosphatases and 142 of their specific regulatory subunits, as well as other regulators of cell proliferation, stress and apoptosis. The detection of specific phosphosites does not only inform on the activity of the relevant kinases but can also provide clues about potential activators.

Huh7.5.1 cells were transfected with Jc1ΔE1E2 transcripts and harvested at given time points, with mock-transfected cells (treated with DMRIE-C only) acting as controls for each time point. Cells were pelleted at 4 °C and shipped to Kinexus for microarray analysis. The analyses were performed following the recommendations from the supplier (www.kinexus.ca). Briefly, 50 μg of lysate protein from each sample were covalently labelled with a different proprietary fluorescent dye. After blocking non-specific binding sites on the array, an incubation chamber was mounted onto the microarray to permit the loading of two samples (normally one control and one matching treated sample) side by side on the same chip and prevent mixing of the samples. Following incubation, unbound proteins were washed away. Each array produces a pair of 16-bit images, which are captured with a Perkin-Elmer ScanArray Reader laser array scanner (Waltham, MA).

Signal quantification was performed using ImaGene 9.0 from BioDiscovery (El Segundo, CA). The background-corrected raw intensity data are logarithmically transformed with base 2. Z scores are calculated by subtracting the overall average intensity of all spots within a sample from the raw intensity for each spot and dividing it by the s.d. of all of the measured intensities within each sample[67]. Z ratios were further calculated by taking the difference between the averages of the observed protein Z scores and dividing by the s.d. of all of the differences for that particular comparison. A Z ratio of 1.2 is inferred as significant[67]. For convenience, the changes in spot intensity between control and treatment samples have been expressed as the percentage of change from control using globally normalized data (Supplementary Data 1–3).

**siRNA silencing.** Gene silencing was performed following standard procedures[68]. The DharmaFECT 4 (DF4) transfection reagent (Dharmacon RNAi Technologies, GE) was used to reverse transfect SMARTpool (pools of four individual siRNAs hybridizing to different sequences in each target transcript) siRNA into 7.5.1 cells in 96-well plate format. siRNA were diluted in 1 × siRNA buffer (Dharmacon) to a final concentration of 4 μM. Cells were trypsinized and resuspended in Dulbecco's minimum essential medium (DMEM) containing 10% FBS (DKSH, Australia). DF4 (0.05 μl per well) was diluted in Opti-MEM (16 μl per well) (Life Technologies) and incubated for 5 min before adding 4 μl of 1 μM SMARTpool

siRNA. The transfection mixtures were incubated at room temperature for further 20 min before 80 µl of cell suspension (5,000 cells) were added to each well. Cells were allowed to settle for 10 min before they were transferred into a 5% $CO_2$ incubator. Twenty-four hours post-transfection, the medium was removed and cells were washed once with DMEM. To each well, 50 µl DMEM plus FBS were added and incubated for a further 12 h. The medium was removed and 40 µl virus stock were added and incubated for 6 h at 37 °C. Cells were washed twice with medium to remove luciferase residual and 70 µl fresh medium was then added. At 15 and 43 h post-infection, 10 µl samples were collected from each well for luciferase assay (described below). HCV321 siRNA (AGGUCUCGUAGACCGUGCA) was used to silence HCV genome as a positive biological control to confirm transfection conditions. The luciferase value of each well (see below) was normalized to cell viability and, subsequently, to ON-TARGETplus non-targeting SMARTpool siRNA (Dharmacon, Cat. No. D-001810-10). The effect of siRNA on viral replication was assessed in three independent experiments, each in duplicate. Discordant values, 50% lower or above the average, were excluded. For each cell factor, the average and s.d. of at least three wells was calculated and expressed as the percentage of non-targeting siRNA (si-OTP-NT). The 103 genes and their specific SMARTpool siRNA sequences are presented in Supplementary Data 5.

**Treatment of cells with a MAP4K2 inhibitor.** The selective MAP4K2 inhibitor, compound 17 (called C17 in the original publication[54], but now renamed TL4-12; see http://graylab.dfci.harvard.edu/index.php?id=61), was a kind gift from Professor N. Gray, Harvard Medical School, Boston, MA. The selectivity and pharmacokinetics of TL4-12 have been fully characterized[54]. The compound was dissolved in DMSO at a final concentration of 10 mM. Cells were grown in 96-well plates ($2 \times 10^4$ cell per well) overnight and treated with twofold dilutions of the inhibitor or the vehicle alone (DMSO) for 5 h. Subsequently, the cells were infected with the virus in the presence of inhibitor. After 6 h incubation with virus, cells in each well were washed twice with 100 µl DMEM and grown in 70 µl fresh medium containing 10% FBS. The first and second 10 µl samples were collected at time points 12 and 36 h, respectively. The luciferase activity in the inhibitor- or mock-treated samples was measured and presented as the percentage of non-treated HCV-infected samples. To determine the effect of TL4-12 on virus replication, cells were treated with the compound or DMSO only (carrier) for 24 h and stained with the HCV NS5A mouse monoclonal antibody 9E10 and Alexa 488-conjugated goat anti-mouse antibody. The signals were visualized using the same exposure time using a Leica laser scanning confocal microscope.

**Western blot analysis.** Huh7.5.1 cells were directly lysed by adding Laemmli buffer (62 mM Tris-HCl, pH 6.8, 25% glycerol, 2% SDS, 0.02% bromophenol blue, 100 mM dithiothreitol) to cell monolayer and denatured at 95 °C for 5 min. Samples were resolved onto a 4–12% polyacrylamide gel (Novex) at 160 V for 55 min and transferred onto a nitrocellulose membrane (BioRad). To detect HCV NS5A, the monoclonal antibody 9E10 was used at a dilution 1/10,000 in blocking buffer (5% skim milk). Bound antibodies were detected using a horseradish peroxidase-conjugated anti-mouse IgG (Cell Signalling Technology, Cat. no. 7076) at a final dilution of 1/2,000. The bound secondary antibody was detected using the Amersham ECL Prime Western Blotting Detection Reagent (GE Healthcares) and the visualized signals were quantified using the Image Lab software (BioRad). Actin was measured as the housekeeping control protein, using a mouse monoclonal antibody (Abcam, Cat. no. ab3280) and the bound antibody was detected by anti-mouse antibody. All signals were normalized to actin. To demonstrate the effect of SMARTpool and individual MAP4K2 siRNA on the MAP4K2 expression levels, $7.5 \times 10^4$ cells per each well of a 24-well plate were reverse transfected as described above (siRNA silencing, Supplementary Data 5). At 24 h post-transfection, the transfection mixture was replaced with fresh medium. The cells were incubated for a further 26 h, and the lysates were prepared and tested as described above. A rabbit MAP4K2 antibody (Cell Signalling Technology, Cat. no. 3282 S) was used at 1/1,000 dilution in the blocking buffer and the bound antibody was detected using a horseradish peroxidase-conjugated anti-rabbit antibody (Cell Signalling Technology, Cat. no. 7074) at a dilution of 1/1,000 in blocking buffer. β-Actin was used as a housekeeping control protein for normalization purposes.

**Luciferase assay.** The Renilla Luciferase Assay Kit (Promega, Madison, WI) was used as per the manufacturer's recommendations. Briefly, working in a Nunc black wall 96-well plate, 10 µl of the collected supernatant fluids were mixed with 10 µl $5 \times$ lysis buffer. To each well, 100 µl substrate were added and the relative light was measured using a BMG FLUOstar OPTIMA Microplate Reader.

**Quantitative reverse-transcription polymerase chain reaction.** Total RNA was extracted with RNeasy Micro Plus Extraction Kit (Qiagen) and eluted in 15 µl RNase-free water. One microlitre of the RNA solution was added to 9 µl of RT–PCR mixture containing oligonucleotide primers and HCV probe (IDT) using a Transcriptor One-Step RT–PCR kit (Roche). Each RNA sample was used in duplicated wells. Following synthesis of viral cDNA for 10 min at 50 °C and transcriptase inactivation for 7 min at 94 °C, 45 thermal cycles (94 °C for 10 s and 60 °C for 1 min) were performed using a Promega Lighcycler machine. The results were normalized against housekeeping genes GAPDH and β-actin[69,70]. To do this, a SuperscriptIII Transcription Single-Strand Synthesis Kit (Invitrogen) and oligo-dT (supplied in the kit) was used to generate cDNA from the gene transcripts using 1 µl total RNA in 10 µl reactions for 30 min at 50 °C. One microlitre of each cDNA was used in duplicated qPCR reactions using a Power SYBR Green PCR Kit (Applied Biosytems). Thermal cycles were performed as described above. The list of oligonucleotides used in the above experiments is provided in Supplementary Data 9.

**Viability assay.** After collecting the final samples for measuring luciferase activity at approximately 90 h post-transfection with siRNA, the viability of cells was tested by PrestoBlue Cell Viability reagent (Thermofisher). Briefly, the $10 \times$ reagent was added to a final $1 \times$ concentration and incubated at 37 °C for 10 min before SDS was added to a final concentration of 1% SDS. Equal amounts of medium containing the above reagents acted as background control. The lysates were collected and the relative fluorescent units were measured using a Tecan Infinite M200 plate reader at excitation and emission 535 and 590 nm, respectively.

**Bioinformatics analyses.** The short-listed proteins and phosphorylation sites from the Kinexus reports were analysed as follows: heatmaps were generated using the heatmap.2 function from the R package gplots. KOBAS[71,72] was used to perform pathway enrichment analysis. The hypergeometric test was selected to test statistical enrichment of KEGG pathways (http://www.genome.jp/kegg/kegg4.html), and the P values were corrected for multiple comparisons using the Benjamini–Hochberg procedure[73]. Published microarray data sets[60] were analysed by the TFactS program[61] using the Sign-sensitive option and the default settings (P-value ≤ 0.05; E-value ≤ 0.05; Q-value ≤ 0.05; false discovery rate ≤ 0.05; random control ≤ 5%, number of random selections = 100) (http://www.tfacts.org/TFactS-new/TFactS-v2/index1.html). The significant values are equal to or lower than the set thresholds and presented in red in the output list (Supplementary Data 8).

**Data availability.** The authors declare that all data supporting the findings of this study are available within the article and its Supplementary Information files or are available from the authors upon request.

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

## Acknowledgements

We thank Professor Charles Rice for providing us with HCV full genome cDNA constructs and antibodies to NS5A. We thank Professor Francis Chisari for the Huh7.5.1 human hepatoma cell line. We thank Professor Nathan Gray and Dr Tan Li (Harvard University) for the TL4-12 compound. We thank Dr Brendan Russ for his assistance with the performance of qRT–PCR. We thank Professor Ralf Bartenschlager (Heidelberg University, Germany) for assisting with control siRNA. Also, we thank Ms Judy Callaghan, the Monash Micro Imaging facility, for their assistance with confocal microscopy. The Victorian Centre for Functional Genomics (K.J.S.) was funded by the

Australian Cancer Research Foundation (ACRF), the Victorian Department of Industry, Innovation and Regional Development (DIIRD), the Australian Phenomics Network (APN) and supported by funding from the Australian Government's Education Investment Fund through the Super Science Initiative, the Australasian Genomics Technologies Association (AGTA), the Brockhoff Foundation and the Peter MacCallum Cancer Centre Foundation. T.F.B. acknowledges grant support of the EU (ERC HEPCAR and H2020 HEPCENT) and NIH (U19-AI123862). This work was made possible through a grant from the Australian Centre for HIV and Hepatitis Virology Research (ACH2).

## Author contributions

G.H. strategically designed and conducted the experimental work and wrote the manuscript with C.D. K.J.S. supervised the siRNA screening experiments, J.W. performed the pathway bioinformatic analyses, H.J.N., R.J.D and T.F.B. provided ideas and contributed to writing the manuscript. C.D. designed the concept, contributed to data analysis and wrote the manuscript with G.H.

## Additional information

**Competing interests:** The authors declare no competing financial interests.

