## [Peer Review File · Nature Communications]

Reviewers' comments:

Reviewer #1 (Remarks to the Author):

This study carries out an antibody microarray to characterise proteins that are up or down regulated at 6h, 12h, and 24h post-HCV transfection, in the Huh7 liver cell line. The first objective of the study was to analyse host cell signaling pathways that are modulated at three time points in order to better understand early changes occurring during HCV replication. Using this approach, the authors show that a number of cellular pathways are altered following infection, which include the NF κ B, JAK/STAT and MAPK pathway. In addition, the technique also highlights alterations in calcium signaling, cell cycle progression and apoptosis. The second objective was to investigate if these altered cellular factors, identified in the microarray, were required for HCV replication. To investigate this, the authors used siRNA gene silencing, followed by a luciferase assay, in order to observe the effect on HCV replication. Overall, this is a detailed comprehensive study that characterizes the early changes and dynamics, which occur following HCV transfection. The study captures these changes at 6h, 12h and 24h in order to compare and contrast alterations in signaling, which could lead to vital clues, regarding how HCV hijacks the cell. Ultimately this method of investigation creates an overall picture that highlights the numerous ways HCV can manipulate the cell to enhance its survival. The importance of this study is underlined by the authors, who say that viral manipulation of signaling could be one of the ways the virus establishes chronic infection, which still devastates thousands of lives. The combination of antibody microarray and siRNA gene silencing is effective method to establish the important factors that are modulated following HCV infection. Using both techniques filters out the factors required for HCV replication and clearly shows which should be a focus of attention. Microarrays generate a large amount of data and the authors effectively and logically organise it. In addition, the study demonstrates a number of novel proteins within very familiar pathways, which will be of interest to researchers in the field.

Major Issues:

1. A huge amount of data is kept in the supplementary figures. It would be beneficial to the paper to consider bringing key discoveries to the main figures (for example in Figure 1b-c it would be beneficial to know which genes are most significantly affected).

Minor suggested improvements:

Abstract corrections:

Line 43-45: The relevance of these findings may indeed have major implications for other pathogens and be therapeutically relevant, but the authors need to explain this more effectively. The current statement is vague.

Introduction corrections:

Line 54: Reference needed

Line 56: Replace "cure" with "sustained virological resistance (SVR)"

Line 68: "Sequential mobilization" is a somewhat misleading phrase, therefore rephrase: "However, there are no reports of comprehensive studies focusing on the sequential mobilization of host cell factors in the early steps of HCV replication."

Line 73: "... the human kinases PAK and MEK1 are activated following infection."

Line 74-78: simplify sentence between lines 74-78

Results and discussion corrections:

Line 96: Full stop needed after respectively

Line 98: Components

Line 102: "Those cell factors that were modulated..."

Line 112: focus

Line 144-145: "NIK was down regulated...and never reached..."

Line 145: Describe the specific "control" here

Line 191-207 JAK/STAT5a section: include brief description of HCV being known to regulate the JAK/STAT pathway through for example Degradation of STAT1 and STAT3 and induction of SOCS3 (including references PMID: 23587486 and PMID: 24659790)

Line 223: Remove "did" at the end of the sentence

Line 233: End the sentence after p53. New sentence "Research has demonstrated that silencing of ..."

Line 254: "...concentrations of TLR4-12 (0.4-6 μ M)..."

Line 274: It is essential to be more specific here, by naming the components

Line 282: whose characterization

Line 289: "... our data indicates.."

For the luciferase experiments, it would be clearer to have labeled on the graph "Relative X luciferase activity"

Reviewer #2 (Remarks to the Author):

In this study, Gholamreza et al. examined HCV-host cell interaction by analyzing the data from antibody microarray. By bioinformatics analysis, they identified genes/signal pathways modulated by HCV RNA transfection. In addition, they studied the effect of the identified genes by using siRNAs or a chemical inhibitor. In this way, the authors tried to study HCV-host cell interaction comprehensively; however, their data need to be further validated as described below.

1. The authors performed HCV RNA transfection to mimic HCV infection. For the validation of the data, it is better to use HCVcc infection systems, particularly for assays with siRNA and TL4-12.

2. The current data from assays with siRNA and TL4-12 rely on luciferase activity. It is better to validate the data by assessing intracellular HCV RNA titers.

3. Assays with siRNA and TL4-12 need to be performed by using various genotypes of HCV RNA in order to know whether the result is limited to certain HCV genotypes or not.

4. The authors suggested that several signal pathways are modulated by HCV RNA transfection. To confirm this result, the authors need to examine mRNA levels of downstream genes for each signal pathway.

5. Assays with TL4-12 need to be performed by using primary human hepatocytes after HCVcc infection.

6. The authors harvested HCV RNA-transfected cells at 6, 12, and 24 hours post-transfection for antibody microarray. However, a longer period is required to examine the effect of HCV in a chronic stage. In fact, HCV infection can be maintained for several days without significant cell death. Therefore, it is better to examine the effect of HCV for a longer period.

7. Minor: Throughout the section of NF- κ B pathway, the reference numbers are incorrect. Please check the references again.

Reviewer #3 (Remarks to the Author):

The work presented here is a very interesting work. The author developed a method to allow the identification of host factors involved in the viral replication of HCV. The interest for such screens is highly increasing, especially to determine host factors to target for drug development in order to impair viral replication and limit/prevent the emergence of resistant strains.

The authors present a very interesting work on the identification of host factors involved in HCV replication. The data obtained with such screen are complicated to interpret and analyze but the authors have done a tedious work to be able to determine the main pathways involved in HCV replication, and then identify within these pathways the host factors having a significant role on HCV infection through gene silencing validation. However, some statements regarding the results of the siRNA validation appear to be

premature and need further statistical analysis.

A weak point of the method is the use of different systems (either replicon or cell culture) depending on the assay (antibody microarray and siRNA silencing). It would have been preferable for the authors to use only the replicon system (Jc1ΔE1E2) for both assays in order to avoid possible effects of the siRNA on other steps of the viral life cycle especially for MAP4K2 that the authors identify as a host factors involved in HCV infection (see major comments c and d). Nonetheless, the method is really interesting and the results are sufficient to highlight the interest of the author's method in the research of host factors involved in viral life cycle.

I- Major comments:

a) The RNAi-based functional validation of the hits identified with the antibody microarray screen needs further statistical analysis.

In the Supplementary Table 6, the authors report the effect of gene silencing on HCV infection using a reporter virus. Figure 1E, they report the top 10 factors of this list, either promoting or inhibiting HCV infection. However, statements regarding the role of the factors for which the gene silencing lead to an increase of infection can't be made without statistical confirmation. Indeed, most of the results for these genes have a wide standard deviation (SD) (5 out of 10 have an SD above 20% of the average value) which overlap with the negative control (at least for 7 out of 10). This could be due to a saturation of the signal during the readout. Simple statistical test could confirm whether their statement is based on statistically significant results.

b) Many statements made by the author seem presumptive regarding the results of the gene silencing validation.

- Lines 129-130: "the 2 NF-κB inhibitors, IκBa and IκBb, reduced virus replication". However, on Figure 2C, there is a clear overlap of the standard deviations from both the negative control si-OTP-NT and the si-NFKBIA (IκBa) and the supplementary table 6 indicates a mean of 80.37 and standard deviation of 37.34, i.e. 46% of the mean value.

- Lines 156 - 157: "Our results indicate IRAK2 has a subtle promoting role in HCV replication". Yet, Figure 2C clearly shows an overlap of the standard deviations from both the negative control si-OTP-NT and the si-IRAK2.

- Lines 204: "silencing of (...) JAK2 (...) resulted in a reduction in virus replication". Again, based on Figure 4C, the effect of the siJAK2 don't appear significant compare to the negative control.

Again, for these statements, a statistical analysis is required

c) The actual effect on the viral replication needs to be strengthen.

Results obtained with the host factors MAP4K2 are really interesting. However, the validation has been performed in the context of the whole infection, using the Jc1-Luc. In this context, the gene silencing could interfere with the viral life cycle by impairing any of its steps, from the entry to the egress. Yet, MAP4K2 seems to have been previously reported to decrease HCVpp infection (Lupberger et al., Nat Med, 2011, supplementary table), which would indicate an effect on the entry. This could also explained the decrease of NS5A observed by both Western Blot and Immunofluorescence. Then, it would be interesting to

assess the effect of MAP4K2 (either by individually transfecting each siMAP4K2, or by using TL4-12) on the Jc1ΔE1E2 transcript.

d) The background of the Western Blot of NS5A after treatment by TL4-12 shows inconsistency.

The effect of TL4-12 is really interesting and strongly highlight the interest of the author's method in the research of host factors involved in viral replication to find and/or develop antiviral therapy. Yet, the western blot appears a bit inconsistent and it would be preferable to do it again.

II- Minor comments:

a) the authors should be more consistent regarding the nomenclature of the genes. Line 129, the authors call the 2 NF-κB inhibitors "IkBa and IkBb", while on Figure 2C, they are called NFKBIA and NFKBIB respectively.

b) Some resistance-associated variants (RAVs) to the DAAs have been reported (Chayama et Hayes, *Viruses*, 2015). If one of such variant is available for in vitro studies and if the authors have access to it, it would be really interesting to assess the effect of such the TL4-12 on RAVs.

c) Regarding the toxicity of the TL4-12 compound, did the authors calculate the selective/therapeutic index?

d) Line 292: a space is required between between "whose" and "characterization"

RESPONSE TO REVIEWERS

We thank the reviewers for their insightful comments and their suggestions, the implementation of which have clearly improved the manuscript. Changes and additions are highlighted in yellow in the manuscript.

The original review is copied below; our response to comments appears in red characters after each comment.

Reviewers' comments:

Reviewer #1 (Remarks to the Author):

This study carries out an antibody microarray to characterise proteins that are up or down regulated at 6h, 12h, and 24h post-HCV transfection, in the Huh7 liver cell line. The first objective of the study was to analyse host cell signaling pathways that are modulated at three time points in order to better understand early changes occurring during HCV replication. Using this approach, the authors show that a number of cellular pathways are altered following infection, which include the NF κ B, JAK/STAT and MAPK pathway. In addition, the technique also highlights alterations in calcium signaling, cell cycle progression and apoptosis. The second objective was to investigate if these altered cellular factors, identified in the microarray, were required for HCV replication. To investigate this, the authors used siRNA gene silencing, followed by a luciferase assay, in order to observe the effect on HCV replication. Overall, this is a detailed comprehensive study that characterizes the early changes and dynamics, which occur following HCV transfection. The study captures these changes at 6h, 12h and 24h in order to compare and contrast alterations in signaling, which could lead to vital clues, regarding how HCV hijacks the cell. Ultimately this method of investigation creates an overall picture that highlights the numerous ways HCV can manipulate the cell to enhance its survival. The importance of this study is underlined by the authors, who say that viral manipulation of signaling could be one of the ways the virus establishes chronic infection, which still devastates thousands of lives. The combination of antibody microarray and siRNA gene silencing is effective method to establish the important factors that are modulated following HCV infection. Using both techniques filters out the factors required for HCV replication and clearly shows which should be a focus of attention. Microarrays generate a large amount of data and the authors effectively and logically organise it. In addition, the study demonstrates a number of novel proteins within very familiar pathways, which will be of interest to researchers in the field.

→ Response: We appreciate the overall positive assessment of the study

Major Issues:

1. A huge amount of data is kept in the supplementary figures. It would benefit the paper to consider bringing key discoveries to the main figures (for example in Figure 1b-c it would be beneficial to know which genes are most significantly affected).

- ➔ Response: We fully agree with the reviewer that the main findings for each pathway would deserve to be included in the main text; however we strived to comply with limits on the size and figure number indicated in the Instructions to Authors, so we had to move data on some of the pathways to Supplementary Information.

Minor suggested improvements:

Abstract corrections:

Line 43-45: The relevance of these findings may indeed have major implications for other pathogens and be therapeutically relevant, but the authors need to explain this more effectively. The current statement is vague.

- ➔ Response: We have modified the sentence as follows: “We propose that this strategy of identifying targets for anti-infective therapeutic intervention within the host cell signalome can be applied to any intracellular pathogen.”

Introduction corrections:

Line 54: Reference needed

- ➔ Response. We inserted a reference to a review covering the subject

Line 56: Replace "cure" with "sustained virological resistance (SVR)"

- ➔ Response: Done

Line 68: "Sequential mobilization" is a somewhat misleading phrase, therefore rephrase: "However, there are no reports of comprehensive studies focusing on the sequential mobilization of host cell factors in the early steps of HCV replication."

- ➔ Response: The sentence now reads as follows: “However, there are no reports of comprehensive studies focusing on time-dependent mobilisation of host cell factors during the early steps of HCV replication.”

Line 73: "... the human kinases PAK and MEK1 are activated following infection."

- ➔ Response: We maintained the original wording –“is” refers to the signalling pathway (“demonstrated that a signalling pathway implicating the human kinases PAK and MEK1 is activated by infection”)

Line 74-78: simplify sentence between lines 74-78

- ➔ Response: We split the sentence, which now reads: “Here, we report that implementation of the Kinexus™ antibody microarray technology coupled to functional validation of hits by siRNA confirmed a number of host cell factors, notably protein kinases, that were previously identified as modulators of HCV infection; this represents a useful positive control for our new approach. Importantly, this also revealed several novel host cell signalling pathways that are mobilised by HCV.”

Results and discussion corrections:

Line 96: Full stop needed after respectively

→ → Done

Line 98: Components

→ → Done

Line 102: "Those cell factors that were modulated..."

→ Response: The sentence now reads "The number of cell factors whose expression or phosphorylation status was modulated by infection"

Line 112: focus

→ → Done

Line 144-145: "NIK was down regulated...and never reached..."

→ Response: The Results are generally described in the present tense; for the sake of internal coherence, we left this sentence in its original form.

Line 145: Describe the specific "control" here

→ Response: The sentence now reads: "...never reaches the same level as the mock-infected cells control throughout the duration of the experiment".

Line 191-207 JAK/STAT5a section: include brief description of HCV being known to regulate the JAK/STAT pathway through for example Degradation of STAT1 and STAT3 and induction of SOCS3 (including references PMID: 23587486 and PMID: 24659790)

→ Response: We have now added the following sentence: "It has been reported that HCV regulates the JAK/STAT pathway, notably through degradation of STAT1 and STAT3 and induction of SOCS3, a member of the suppressors of cytokine signalling (SOCS) family that inhibit STAT phosphorylation by binding and inhibiting JAKs [two new references inserted here: PMID: 23587486 and PMID: 24659790]. We found that the phosphorylated..."

Line 223: Remove "did" at the end of the sentence

→ → Done

Line 233: End the sentence after p53. New sentence "Research has demonstrated that silencing of ..."

→ Response: Done. The sentence now reads “and is a transcriptional target of p53. It has been reported that silencing PAC1 leads to the suppression of apoptosis and promotes cell survival”.

Line 254: “..concentrations of TLR4-12 (0.4-6µM)...”

→ → Done

Line 274: It is essential to be more specific here, by naming the components

→ Response: The components are named in immediately subsequent sentences relating to promoters or suppressors of infection.

Line 282: whose characterization

→ → Done

Line 289: “... our data indicates...”

→ → Done

For the luciferase experiments, it would be clearer to have labeled on the graph “Relative X Luciferase activity”

→ Response: We do not understand the request from the reviewer, but would be happy to receive a suggestion to render the labelling clearer.

Reviewer #2 (Remarks to the Author):

In this study, Gholamreza et al. examined HCV-host cell interaction by analyzing the data from antibody microarray. By bioinformatics analysis, they identified genes/signal pathways modulated by HCV RNA transfection. In addition, they studied the effect of the identified genes by using siRNAs or a chemical inhibitor. In this way, the authors tried to study HCV-host cell interaction comprehensively; however, their data need to be further validated as described below.

1. The authors performed HCV RNA transfection to mimic HCV infection. For the validation of the data, it is better to use HCVcc infection systems, particularly for assays with siRNA and TL4-12.

→ Response: The siRNA and TL4-12 experiments have actually been performed using the HCVcc infection system, not the RNA transfection. After transfection of the siRNA (which may have led to the confusion), the cells were infected (not transfected) with virus. We double-checked that this is correctly stated in the Materials and Methods section.

2. The current data from assays with siRNA and TL4-12 rely on Luciferase activity. It is better to validate the data by assessing intracellular HCV RNA titers.

→ Response: Contaminant input viral RNA is a major issue in using RNA titre to monitor virus production, and it is well accepted to use a reporter gene to measure viral replication, as reported (for example) in references 21-23. We nevertheless repeated the siRNA screen for all genes presented in Fig 2-5, and used qRT-PCR to quantify RNA titers. Additionally, the effects of MAP4K2 individual siRNA and TL4-12 on virus replication were measured by qRT-

PCR (Fig 6B and D). Overall there was good agreement between the luciferase and qRT-PCR assays, as expected. This dual strategy is briefly described at the end of the first Results section (lines 114-124), which now reads:

→ “The modulation of HCV replication following silencing of cell factors was measured using a reporter HCV cell culture system by the measurement of a secreted luciferase enzyme encoded by a modified viral genome, which is an accepted quantifier of virus replication (references 21-23). The results were normalised against cell viability and are presented in Supplementary Table 6. The primary screen was validated by direct, qRT-PCR quantification of intracellular viral genomic RNA levels following infection with the virus; in this case, the results were normalised against mRNA levels of the housekeeping genes β -actin and GAPDH and are presented in Supplementary Table 7. The combination of the two systems provided strong evidence for the implication of several cell factors in HCV replication. For a few tested factors, some discrepancy between the results of the luciferase and qRT-PCR readout systems was observed (see Discussion).” qRT-PCR data and luciferase data are presented together in each of the Figures 2 to 5.

3. Assays with siRNA and TL4-12 need to be performed by using various genotypes of HCV RNA in order to know whether the result is limited to certain HCV genotypes or not.

→ Response: We fully concur with the reviewer regarding the interest of investigating other HCV genotypes. The main thrust of the manuscript is to provide a proof of concept that the combination of antibody microarray and gene silencing approaches represented a powerful novel strategy to identify potential therapeutic targets. While it would definitely be of great interest to compare various HCV genotypes in this context (and hence possibly provide explanations for phenotypic differences), we feel that this lies outside the scope of the current manuscript.

4. The authors suggested that several signal pathways are modulated by HCV RNA transfection. To confirm this result, the authors need to examine mRNA levels of downstream genes for each signal pathway. The components of signalling pathways may modulate viral replication independently of their final transcription factors.

→ Response: This is an excellent point. A transcriptomics study of HCV infection has been published, based on the same time points as those in our own study; we have mined the relevant datasets, and added to the Discussion section a paragraph elaborating on these in relation to our antibody microarray data (Lines 343-376). Specifically, we analysed the transcriptomics datasets at the three time points using the TFactS program, which predicts which transcription factors are likely to be activated or inhibited based on transcriptomics datasets. As detailed in the manuscript, this showed good agreement with some of the pathways we detected.

5. Assays with TL4-12 need to be performed by using primary human hepatocytes after HCVcc infection.

→ Response: We agree that this would be ideal (and required) in the context of a drug discovery program centered on TL4-12, and this will be part of our follow-up studies. However, we feel that a clear effect in cell lines is sufficient to establish that the compound

interferes with viral production. Furthermore, we now provide additional data on the Mode of Action of the compound (see below).

6. The authors harvested HCV RNA-transfected cells at 6, 12, and 24 hours post-transfection for antibody microarray. However, a longer period is required to examine the effect of HCV in a chronic stage. In fact, HCV infection can be maintained for several days without significant cell death. Therefore, it is better to examine the effect of HCV for a longer period.

→ **Response: The major aim of this study is to uncover host signalling response to early stages of HCV infection. We fully agree that investigating chronic infection from a signalling perspective would be extremely interesting. However, we feel that this lies outside the scope of this report.**

7. Minor: Throughout the section of NF- κ B pathway, the reference numbers are incorrect. Please check the references again.

→ **Response: All references have been checked**

Reviewer #3 (Remarks to the Author):

The work presented here is a very interesting work. The author developed a method to allow the identification of host factors involved in the viral replication of HCV. The interest for such screens is highly increasing, especially to determine host factors to target for drug development in order to impair viral replication and limit/prevent the emergence of resistant strains.

The authors present a very interesting work on the identification of host factors involved in HCV replication. The data obtained with such screen are complicated to interpret and analyze but the authors have done a tedious work to be able to determine the main pathways involved in HCV replication, and then identify within these pathways the host factors having a significant role on HCV infection through gene silencing validation. However, some statements regarding the results of the siRNA validation appear to be premature and need further statistical analysis.

A weak point of the method is the use of different systems (either replicon or cell culture) depending on the assay (antibody microarray and siRNA silencing). It would have been preferable for the authors to use only the replicon system (Jc1 Δ E1E2) for both assays in order to avoid possible effects of the siRNA on other steps of the viral life cycle especially for MAP4K2 that the authors identify as a host factors involved in HCV infection (see major comments c and d). Nonetheless, the method is really interesting and the results are sufficient to highlight the interest of the author's method in the research of host factors involved in viral life cycle.

→ **Response: We appreciate the overall positive assessment of the study**

I- Major comments:

a) The RNAi-based functional validation of the hits identified with the antibody microarray screen needs further statistical analysis.

In the Supplementary Table 6, the authors report the effect of gene silencing on HCV infection using a reporter virus. Figure 1E, they report the top 10 factors of this list, either promoting or inhibiting HCV infection. However, statements regarding the role of the factors for which the gene silencing

lead to an increase of infection can't be made without statistical confirmation. Indeed, most of the results for these genes have a wide standard deviation (SD) (5 out of 10 have an SD above 20% of the average value) which overlap with the negative control (at least for 7 out of 10). This could be due to a saturation of the signal during the readout. Simple statistical test could confirm whether their statement is based on statistically significant results.

→ **Response: We have now provided statistical analysis of the data in Supplementary Table 6. We have used the unequal variance t-test (available in Excel) and added the obtained p values in a separate column. The effects of siRNA on viral replication was assessed in three independent experiments, each in duplicate. Discordant values, 50% lower or above the average, were excluded, which explains small discrepancies with the dataset in the original submission, such as the IRAK2 si-RNA effect (80% versus 60%).**

b) Many statements made by the author seem presumptive regarding the results of the gene silencing validation.

- Lines 129-130: "the 2 NF- κ B inhibitors, I κ Ba and I κ B β , reduced virus replication". However, on Figure 2C, there is a clear overlap of the standard deviations from both the negative control si-OTP-NT and the si-NFKBIA (I κ Ba) and the supplementary table 6 indicates a mean of 80.37 and standard deviation of 37.34, i.e. 46% of the mean value.

- Lines 156 - 157: "Our results indicate IRAK2 has a subtle promoting role in HCV replication". Yet, Figure 2C clearly shows an overlap of the standard deviations from both the negative control si-OTP-NT and the si-IRAK2.

- Lines 204: "silencing of (...) JAK2 (...) resulted in a reduction in virus replication". Again, based on Figure 4C, the effect of the siJAK2 don't appear significant compare to the negative control. Again, for these statements, a statistical analysis is required

→ **Response: See response to comment above**

c) The actual effect on the viral replication needs to be strengthen.

Results obtained with the host factors MAP4K2 are really interesting. However, the validation has been performed in the context of the whole infection, using the Jc1-Luc. In this context, the gene silencing could interfere with the viral life cycle by impairing any of its steps, from the entry to the egress. Yet, MAP4K2 seems to have been previously reported to decrease HCVpp infection (Lupberger et al., Nat Med, 2011, supplementary table), which would indicate an effect on the entry. This could also explained the decrease of NS5A observed by both Western Blot and Immunofluorescence. Then, it would be interesting to assess the effect of MAP4K2 (either by individually transfecting each siMAP4K2, or by using TL4-12) on the Jc1 Δ E1E2 transcript.

→ **Response: This is an excellent point. We have now added a figure (Fig. 7C) showing the effect of the genes silencing and TL4-12 treatment. We generated an HCV replicon lacking the E1 and E2 proteins, and used it to demonstrate that viral RNA replication is affected when MAP4K2 is silenced or inhibited; this is represented in a new panel in Fig.6. Results from these experiments show that MAP4K2 gene silencing and chemical inhibition suppresses viral genome replication.**

d) The background of the Western Blot of NS5A after treatment by TL4-12 shows inconsistency. The effect of TL4-12 is really interesting and strongly highlight the interest of the author's method in

the research of host factors involved in viral replication to find and/or develop antiviral therapy. Yet, the western blot appears a bit inconsistent and it would be preferable to do it again.

→ **Response: We have carefully assessed the blot, and we are not sure what the exact issue is that the Reviewer has raised.**

II- Minor comments:

a) the authors should be more consistent regarding the nomenclature of the genes. Line 129, the authors call the 2 NF- κ B inhibitors "IkBa and IkBb", while on Figure 2C, they are called NFKBIA and NFKBIB respectively.

→ **Response: The figure labelling was amended. Please note that "NFKBIA" and "NFKBIB" is the nomenclature used in the Kinexus microarray –this has been clarified in the Figure legend.**

b) Some resistance-associated variants (RAVs) to the DAAs have been reported (Chayama et Hayes, Viruses, 2015). If one of such variant is available for in vitro studies and if the authors have access to it, it would be really interesting to assess the effect of such the TL4-12 on RAVs.

→ **Response: This is an excellent suggestion, and we intend to include this in follow-up studies on MAP4K2.TL4-12; however, we feel this lies outside the scope of the present study.**

c) Regarding the toxicity of the TL4-12 compound, did the authors calculate the selective/therapeutic index?

→ **Response: The selectivity and pharmacokinetics of the compound have been published in ref 52. This is now mentioned in the text. We also included the following sentence: "(we used of the Graphpad program to calculate that the concentrations causing 50% toxicity and 50% HCV replication inhibition were 26.8 μ M and μ M 3.9, respectively, giving an approximate therapeutic index of 6.8)."**

d) Line 292: a space is required between between "whose" and "characterization"

→ **→ Done**

REVIEWERS' COMMENTS:

Reviewer #2 (Remarks to the Author):

The authors appropriately revised the manuscript according to reviewers' comments.

Reviewer #3 (Remarks to the Author):

By using an RNAi-based analysis, the authors have been able to identify overall mechanisms and pathways involved in the host response, providing an interesting comprehensive picture of the modulation of host signaling pathways upon HCV infection. Moreover, using this method, they identified specific factors and assessed their role in the viral life cycle. This work is a real asset for both comprehensive knowledge and identification of target for drug development, increasing the interest for such method.

Finally, the authors have properly answered the questions previously addressed and performed the experiments required, strengthening the paper that way.

Minor comment:

line 376: "allowed" is misspelled (alowed)

Response to reviewers:

Query from Reviewer 3

line 376: "allowed" is misspelled (aloowed)

→ corrected